# The Spatial Equity of Nursing Homes in Changchun: A Multi-Trip Modes Analysis

**Shuju Hu [1], Wei Song [2,\*], Chenggu Li [1] and Jia Lu [3]**

[1]  School of Geographical Sciences, Northeast Normal University, Changchun, Jilin 130024, China; husj163@nenu.edu.cn (S.H.); lcg6010@nenu.edu.cn (C.L.)

[2]  Department of Geography and Geosciences, University of Louisville, Louisville, KY 40292, USA

[3]  Geosciences Program, Valdosta State University, Valdosta, GA 31698, USA; jlu@valdosta.edu

\*  Correspondence: wei.song@louisville.edu; Tel.: +1-502-852-2690

**Abstract:** Based on network analysis, different trip modes were integrated into an improved potential model, and the geography of the spatial equity of nursing homes in Changchun is explored in 5-min, 10-min and 15-min scenarios, respectively. Results show that: (1) trip modes have significant influence on spatial equity and that the geography of spatial equity varied with trip modes; (2) the spatial equity value in Changchun is overall kept to a very low level. Most areas in urban fringes and urban core areas belong to underserved areas, and the capacity of nursing home, travel cost and the number of seniors, are the main influencing factors; (3) the geography of spatial equity in different scenarios show a very similar ring structure; namely, the spatial equity value within the urban core and at the most urban periphery is lower than that in intermediate areas. The hot spot analysis showed that the southwest urban fringes and east of the urban core are hot spot areas, while the urban core itself has cold spot areas.

**Keywords:** network analysis; nursing homes; trip mode; accessibility; spatial equity

## 1. Research Background and Literature Review

Currently, China is in a period of rapid demographic, social, and economic transformation. The crisis of an aging population is worsening. In 2017, the number of elderly people in China (over 60 years of age) reached 240 million [1,2]. This makes China the country with the largest number of elderly people in the world [3]. Changchun is the capital city of Jilin province, located in the plains of Northeast China, which is currently growing into an aging city. The city's average aging population ratio has grown from 8.32% to 13.64% from 2010 to 2016. There is no doubt that nursing homes will become increasingly important urban service facilities. Differing from the usual western definition, nursing homes in this paper mainly refer to day care centers for the elderly, which can provide temporary daytime care, accommodation, health care, rehabilitation and amenities for elderly people. Thus, the accessibility of nursing homes and transportation costs associated with nursing homes is very important. In recent years, the Changchun municipal government is making a spatial plan for nursing homes, where to place the facilities for the elderly has also become a topic of common concern for scholars and policy makers. In this context, it is increasingly urgent and necessary to explore the spatial equity issue of nursing homes in Changchun.

Spatial equity usually refers to different residents (regardless of their social class, income, or race) having equal access to certain services [4–9]. Early scholars believed that spatial equity only meant the uniform spatial distribution of service facilities or ensuring the equality of distance to service facilities. Particular attention was given to spatial dimensions [5,8,10–12]. After that, however, the definition of spatial equity was expanded into social dimensions, such as class, income, and age [13]. For example, Talen and Anselin (1998) believed that spatial equity refers to the coordination of facilities or services between different socioeconomic groups [8]. Omer (2006) believed that spatial equity refers to the degree to which the services or facilities were equally distributed in different economic, ethnic, or age groups [7]. In recent years, the integration of different trip modes into traditional accessibility models has become a burgeoning field. Some scholars believe that trip modes have significant influence on spatial equity of service facilities, because different modes of transport can produce different accessibility landscapes [14–16] What is more, integrating trip modes into traditional accessibility models can better reflect different social groups' ability to access certain services [17].

Currently, the metrics measuring spatial equity mainly include the accessibility-based approaches [7–9,18,19], the Lorentz curve method [20], the Gini coefficient method [21], the coefficient of variation (CV) [22], and the Teil index [23]. Among them, the accessibility-based approaches are the most commonly used methods in the assessment of spatial equity of service facilities [12,24–26] and are widely used in the evaluation of spatial equity of urban green spaces [18,27], public playgrounds [28], educational resources [29], public transport [14], healthcare facilities [30], and so forth. Among the accessibility-based approaches, the gravitational potential model, two-step floating catchment method (2SFCA) and their corresponding improved models are relatively popular methods for measuring spatial equity. For example, based on gravity model, Chang (2011) developed a spatial equity index to explore the spatial equities of urban public facilities from both accessibility and mobility perspectives [25]. Talen (1998) evaluated the spatial equity of public playgrounds by a gravity potential model [8]. Vadrevu (2016) assessed the spatial equity of maternal health services by an enhanced two-step floating catchment area method [22]. Shen (2017) explored the spatial equity of public green space among different resident groups by a Gaussian-based two-step floating catchment area method [31]. Dadashpoor (2016) developed an integrated index of spatial equity to assess the spatial inequity of urban facilities in a disaggregated and aggregated manner [32]. Almohamad (2018) assessed the spatial equity of public green spaces by average nearest neighbor and network analysis [18]. Taleai (2014) developed a Spatial Multi-Criteria Analysis (SMCA) method to assess the spatial equity of urban public facilities at different spatial scales [33].

Distance is one of the keys in any methodology of accessibility. Network distance has been proven to be more accurate and more realistic than Euclidean distance with accessibility [34]. As shown in Figure 1, if A is a spatial unit and B is a facility, the red line is the network distance, the blue line is the linear distance, and the radius of the circle is the distance threshold judging whether or not a facility is accessible. If measured by linear distance, B is accessible to A. If calculated by network distance, however, B is not accessible to A. Additionally, trip modes also have an important impact on the measurement of accessibility [6,14,16], and accessibility landscape also varies with different trip modes. The accessibility range of different trip modes in the same time are significantly different from each other. Therefore, integrating trip modes into the measurement of accessibility is very significant. Threshold is another key to accessibility research; most scholars set a time threshold between 5-min to 30-min, but when taking the walking mode into consideration, 5-min, 10-min, and 15-min are typical time divisions [16,19,35–37], and the speeds of walking, public transport, and car in urban contexts are normally set to 5km/h, 25km/h, and 30km/h, respectively [14,16,37].

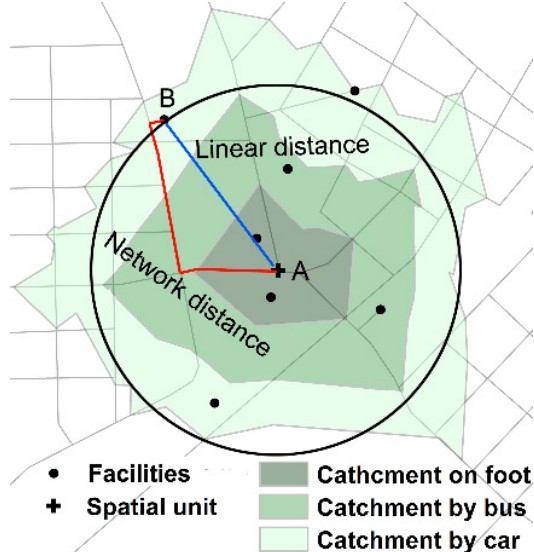

**Figure 1.** The influence of distance and trip modes on accessibility.

Some scholars measured the spatial equity of service facilities by integrating trip modes into traditional accessibility models. For example, in order to explore the influence of trip modes on the spatial equity of healthcare facilities, Mao and Nekorchuk (2013) proposed a Two-Step Floating Catchment Area Method (2SFCAM) and integrated bus and car modes into a Two-Step Floating Catchment Area (2SFCA) model [38]. Shen and Sanchez (2005), as well as Chang and Liao (2011) considered the impact of walking and driving on spatial equity and integrated those trip modes into a potential model [6,39]. Coline (2015) also developed a Variable-Width Floating Catchment Area (VFCA) method and assessed accessibility to urban parks in four trip modes: bicycling, driving, public transit, and walking [15]. Mitchel Langford (2015) claimed that combining public and private transport modes into traditional 2SFCA methodology could contribute to accurately allocating services among different social groups [17]. Xing and Liu (2018) developed a multi-mode 2SFCA and measured spatial disparity of urban parks in three trip modes: walking, cycling and driving, and claimed a multi-mode model can better reflect accessibility [40]. The studies conducted in the multi-mode environment show that the accessibility models integrating trip modes can better reflect social equity [15,17], and that a traditional non-mode model overestimates accessibility. Therefore, the multi-modes model can provide a more realistic evaluation and offers a better guidance for practices [38,40].

In this paper, based on network analysis, the trip modes of walking, bus, and car were integrated into the improved potential model and the spatial equity of nursing homes in Changchun are explored in 5-min, 10-min, and 15-min scenarios, respectively. The main research objectives of this paper are as follows: What is the geography of spatial equity under different trip modes and different scenarios? Will the geography of spatial equity in different scenarios show the same spatial pattern?

The remainder of this paper is organized as follows: Section 2 provides an introduction to the data sources, as well as our research methodology. Section 3 includes analysis of the spatial differences of spatial equity of nursing homes in different scenarios and its influencing factors and spatial patterns. Finally, the key conclusions and the theoretical and policy implications of this research are outlined in Section 4.

## 2. Data Sources and Research Methods

### 2.1. Data Sources

The study area is Changchun's urban district, comprising 58 subdistricts, with the geographical center of each subdistrict abstracted as a demand point. The data of nursing homes (included each nursing home's name, capacity, location, amenities equipped, etc.) were collected from the Changchun

Civil Affairs Bureau. The population of the elderly in 2015 was collected from the Changchun Bureau of Statistics and the road network data of 2018 were provided by the Changchun Transport Agency. As shown in Figure 2a, the network analysis data set includes the road network, the 171 nursing homes, the 58 subdistricts, and the 4062 network junctions. The sub-catchments under different trip modes in each scenario are created by the module of "Network analysis > New service area" in the Arcgis10.4 software. The origins are the subdistricts and the destinations are nursing homes. The number of beds and the number of amenities (physical therapy equipment, fitness and recreational facilities) equipped in each nursing home, as well as the number of scenarios in each subdistrict are shown in Figure 2b. We can see that large nursing homes tend to be located in the intermediate region between the urban core and the urban fringe.

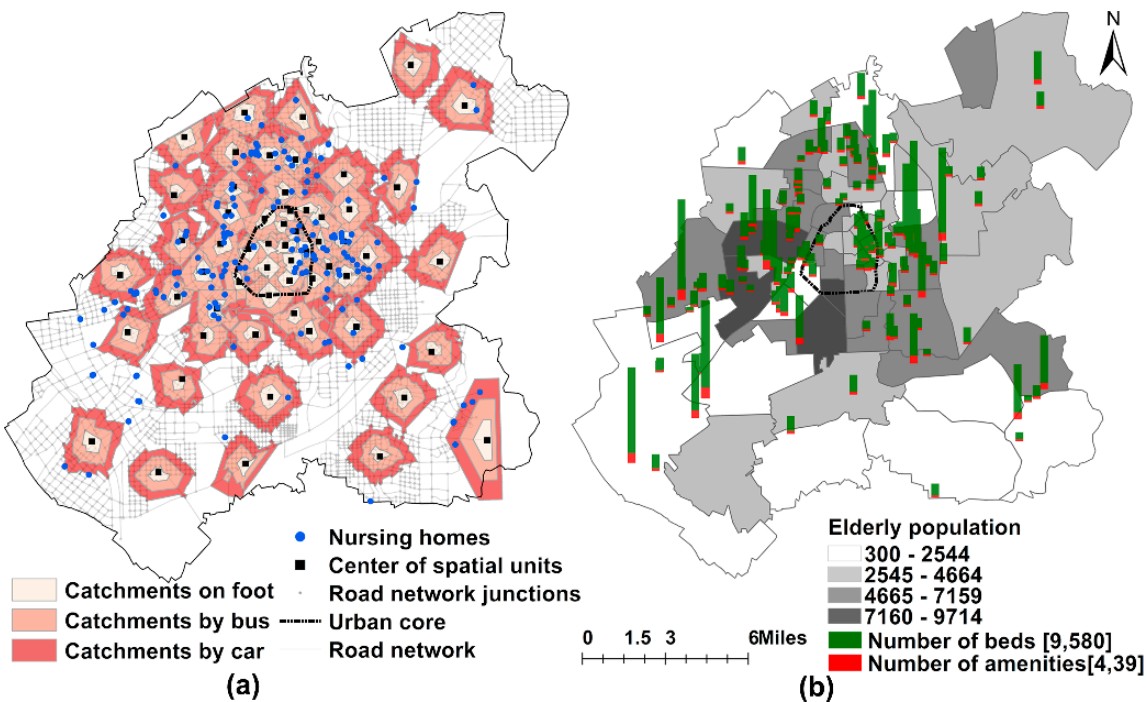

**Figure 2.** The network analysis data set of nursing homes in Changchun. (**a**) the road network and locations of nursing homes; (**b**) the spatial distribution of elderly population and the capacity of nursing homes.

## 2.2. Research Methods

### 2.2.1. Improved Potential Model

The general formula of the gravitational potential model by Hansen in 1959 [41] is as follows:

$$A_i = \sum_{j=1}^{n} \frac{M_j}{D_{ij}^{\beta}} \tag{1}$$

Due to the fact that the gravitational potential model does not take into account the competition for limited resources, Joseph (1982) developed a modified gravitational potential model [42]. The formula of the modified potential model is as follows:

$$A_i = \sum_{j=1}^{n} \frac{M_j}{V_j D_{ij}^{\beta}} \text{ , and } V_j = \sum_{k=1}^{n} \frac{P_k}{D_{kj}^{\beta}} \tag{2}$$

where, $A_i$ represents the accessibility of certain facility in $i$th spatial unit, $V_j$ represents the competition among people in $i$th spatial unit and its neighboring spatial units for the limited services. where, the competition varies proportionally to the population, but inversely with distance. $P_k$ is the total population in nearby spatial unit $k$; $M_j$ is the service capability of facility $j$; $D_{ij}$ is the spatial separation (time or distance) between spatial $i$ and facility $j$; $D_{kj}$ is the spatial separation between spatial unit $k$ and spatial unit $j$; $\beta$ is the friction coefficient. The difference between the improved gravitational potential model and the traditional gravitational potential model is whether or not they take the competition among $i$th spatial unit and other spatial units for the limited resources into account. The competition factor ($V_j$) matters, because consumers do not necessarily choose nearby services; rather, they may also seek services from outer regions. Therefore, the more people that the nearby spatial units have, the fiercer the competitions for the limited resources.

### 2.2.2. Spatial Equity Model

Based on the general gravitational potential model, Chang (2011) and Liang (2013) developed the spatial equity mode by integrating different modes of transport into the general gravitational potential model [6,38]. In this paper, based on the spatial equity mode developed by Chang (2011), we try to further improve this model by integrating competition factor ($V_j$) and the attraction factor ($S_j$) into the spatial equity model. Spatial separation or travel costs were calculated by formula (3),

$$Travel\ cost = \sum_i \left( a_1 \times p_i \times T_{i(walk)} + a_2 \times p_i \times T_{i(car)} + a_3 \times p_i \times T_{i(bus)} \right) \tag{3}$$

where, $p_i$ is the amount of elderly population in subdistrict $i$ ($i$ = 1, 2, 3, ... , 58), and $a_1$, $a_2$, $a_3$ are the proportions of seniors that travel by walking, by car and by public transport in subdistrict $i$, respectively. $a_i \times p_i$ is the number of seniors who travel by each trip mode in subdistrict $i$ and $T_{i(walk)}$, $T_{i(car)}$, $T_{i(bus)}$ are the average travel time from subdistrict $i$ to accessible facilities by walking, car and bus, respectively. Travel cost was then integrated into the improved potential model formula (2) to replace the $D_{ij}^{\beta}$. Thus, the spatial equity of nursing homes can be calculated under a multi trip modes context according to the following, formula (4):

$$SE_i = \frac{\sum_j M_j \times S_j}{V_j \times \sum_i a_1 \times p_i \times T_{i(walk)}} + \frac{\sum_j M_j \times S_j}{V_j \times \sum_i a_2 \times p_i \times T_{i(car)}} + \frac{\sum_j M_j \times S_j}{V_j \times \sum_i a_3 \times p_i \times T_{i(bus)}} \tag{4}$$

where, $SE_i$ is the spatial equity value of nursing homes of subdistrict $i$, and $M_j$ is the service capability of nursing home $j$ falling within each sub-catchment of subdistrict $i$ under different trip modes, which was represented by its total number of beds. $S_j$ is normalized attraction coefficient, which was represented by the number of amenities (physical therapy equipment, fitness and recreational facilities) equipped in each nursing home. $S_j$ was processed by min–max normalization method and its value between 0 and 1, a higher value meaning more attraction. Based on the resident survey of trip modes released by the Changchun Planning Bureau in 2018 [43], the proportions of people travelling by walking, public transport, private cars, and taxi in that year were 30.2%, 19.0%, 20.3% and 19.4%, respectively. The latter two trip modes were combined into one category, namely car mode. Therefore, $a_1$ = 0.302, $a_2$ = 0.397 and $a_3$ = 0.19. A higher $SE_i$ implies a higher accessibility of nursing home services.

As shown in Figure 3, the analysis process can be divided into four steps. Step 1: Based on network analysis and the preset speeds of different trip modes, setting sub-catchments under a different trip mode (walk, bus, and car) for subdistrict $i$ in each scenario. Step 2: Calculating the total service capacities of nursing homes falling within each sub-catchment under different trip modes; calculating the total travel costs from subdistrict $i$ to each nursing home falling within each sub-catchment under different trip modes; calculating the spatial competition ($V_j$) between subdistrict $i$ and other subdistricts. Step 3: Calculating the sub-scores of $SE_m$ under different trip modes in each scenario. Step 4: Summing up the sub-scores of $SE_i$ under different trip modes in each scenario.

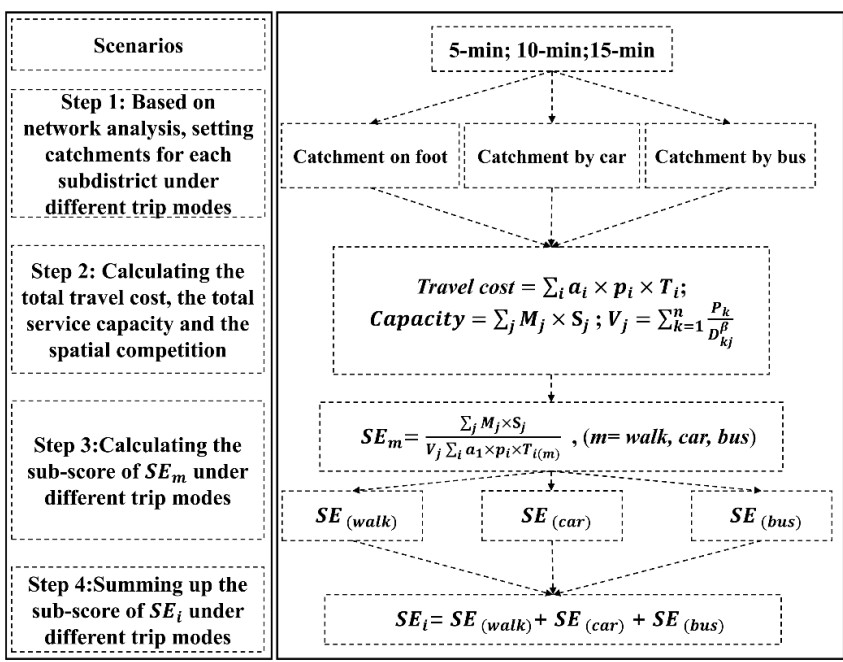

**Figure 3.** The technical details and analysis steps.

### 2.2.3. Hot Spot Analysis

*Getis-Ord Gi\** is a widely used statistic in Arcgis software for detecting the pattern of spatial data; the formula of *Getis-Ord Gi\** statistic is as follows:

$$G_i^* = \frac{\sum_{j=1}^n w_{i,j} x_j - \overline{X} \sum_{j=1}^n w_{i,j}}{S \sqrt{\frac{\left[n \sum_{j=1}^n w_{i,j}^2 - \left(\sum_{j=1}^n w_{i,j}\right)^2\right]}{n-1}}} \text{ and } \overline{X} = \frac{\sum_{j=1}^n x_j}{n}, \ S = \sqrt{\frac{\sum_{j=1}^n x_j^2}{n} - \left(\overline{X}\right)^2} \tag{5}$$

where $x_j$ is the attribute value of spatial unit $j$, $w_{i,j}$ is the spatial weight between spatial unit $i$ and $j$, $n$ is the total number of subdistricts. The spatial patterns of spatial data can be classified into hot spot, cold spot and not significant. Hot spot is a spatial unit with a high value and is surrounded by high value neighbors. Cold spot is a spatial unit with a low value and is surrounded by low value neighbors.

## 3. Results

### 3.1. The Geography of Spatial Equity in Different Scenarios

As shown in Figure 4, the $SE_i$ of nursing homes was calculated under different trip modes in 5-min, 10-min and 15-min scenarios, respectively. The $SE_i$ in each scenario was classified into three grades, namely, high, middle, and low. The geography of $SE_i$ under different trip modes is totally different in the same scenario. Overall, the $SE_i$ in car mode is significantly higher than that in bus and walking modes in each scenario. The $SE_i$ in walking mode is extremely low. In the 5-min scenario, the mean value of $SE_i$ of 58 subdistricts in walking mode is merely 0.64. By contrast, the mean values of $SE_i$ in public transit and car modes are 18.5 and 19.9, respectively. In the 10-min scenario, the spatial equity value in walking mode is still at a very low level (1.02), while the average $SE_i$ in public transit and car modes increases rapidly to 32.5 and 41.3, respectively. In the 15-min scenario, the mean values of $SE_i$ of 58 subdistricts under public transit and car modes further increase to 53.6 and 73.6, respectively, while the $SE_i$ in walking mode is 1.32. Although $SE_i$ increases with time in the same trip mode, the growth rate of $SE_i$ under public transit and car modes are higher than that in walking mode. The average growth rates of $SE_i$ in each 5-min to 15-min interval under walking, public transit, and car mode are 44%, 70%, and 93%, respectively.

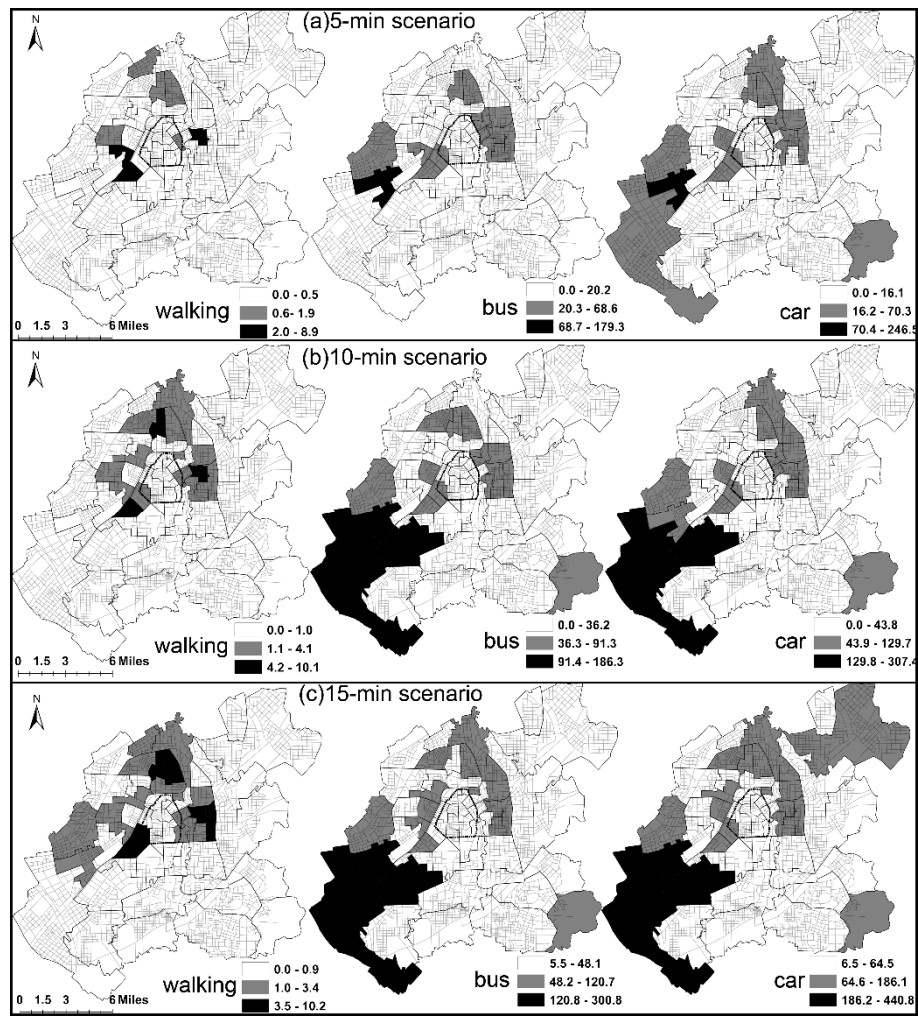

**Figure 4.** The geography of $SE_i$ under different trip modes. (**a**) 5-min scenario; (**b**) 10-min scenario; (**c**) 15-min scenario.

The integrated maps and the descriptive statistical indicators of $SE_i$ in different scenarios in Changchun are shown in Figure 5. and Table 1. In the 5-min integrated scenario, the mean value of spatial equity of 58 subdistricts is 40.2 and the SD is 61.7. There are 40 subdistricts (69%) belonging to underserved areas, and the areas with low $SE_i$ are mainly located in urban core areas and most urban fringes. These areas are at a disadvantage in accessing nursing homes over other regions. In the 10-min integrated scenario, the mean value and SD of $SE_i$ of 58 subdistricts increase to 81.9 and 96.2, respectively. The number of underserved subdistricts reduces to 35; in other words, even in the 10-min scenario, almost 60% of the subdistricts of Changchun are still short of nursing home resources. In the 15-min integrated scenarios, the mean value and SD of 58 subdistricts further increase to 130.9 and 145.7, respectively. More than half of the subdistricts (52%) still belong to underserved areas. Overall, we can conclude that the $SE_i$ of nursing homes in Changchun is varied with trip modes and spatial scales. The $SE_i$ under car mode is significantly higher than that under bus and walking modes in the same scenario. With the increase of time threshold, the mean value and SD of $SE_i$ keep rising, and the spatial variance of nursing home resources among the 58 subdistricts increases. The areas with low $SE_i$ are mainly located in the urban core areas and most urban fringes, while the areas with relative higher $SE_i$ are located in the southwest and southeast urban fringes, as well as the intermediate areas between the urban core and the urban fringes.

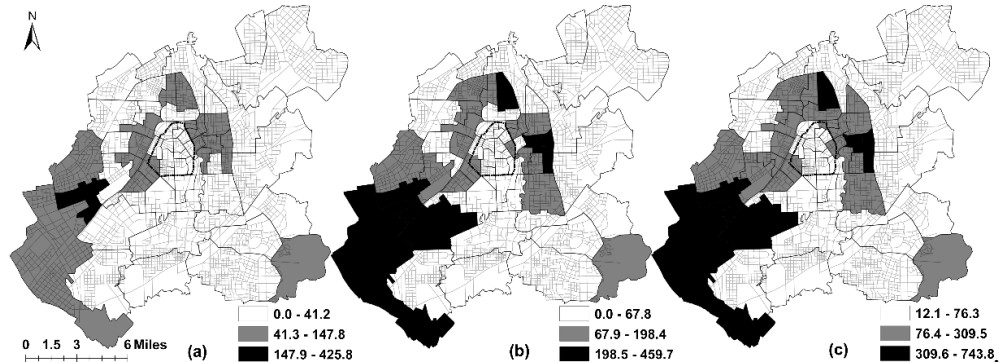

**Figure 5.** The geography of $SE_i$ in different integrated scenarios. (**a**) 5-min integrated scenario; (**b**) 10-min integrated scenario; (**c**) 15-min integrated scenario.

**Table 1.** The descriptive statistical indicators of $SE_i$ in different scenarios.

| Scenarios | $SE_i$ | | | |
|---|---|---|---|---|
| | Maximum | Minimum | Mean | SD |
| 5-min | 425.8 | 0 | 40.2 | 61.7 |
| 10-min | 459.7 | 0 | 81.9 | 96.2 |
| 15-min | 743.8 | 11.9 | 130.9 | 145.7 |

*3.2. The Influencing Factors of Spatial Equity*

To analyze the factors influencing spatial equity of nursing homes in Changchun, we chose the total number of beds ($X_1$) and the total number of nursing homes ($X_4$) falling within the catchment area of $i$th subdistrict, the number of elderly people ($X_2$), the total travel time ($X_3$) and the competition factor $V_j$ ($X_5$) of $i$th subdistrict as the independent variables, and chose $SE_i$ ($Y$) of each subdistrict as the dependent variable. The regression equations in different scenarios are established by stepwise method in SPSS 20.1 software. The equations in different scenarios are as follows:

$$Y_{5\text{-}min} = 60.109 + 0.045X_1 - 0.010X_2 + 2.735X_3; \tag{6}$$

$$Y_{10\text{-}min} = 124.053 + 0.054X_1 - 0.018X_2 + 2.743X_3; \tag{7}$$

$$Y_{15\text{-}min} = 208.239 + 0.073X_1 - 0.034X_2 + 2.924X_3 - 15036.26X_5; \tag{8}$$

In Equations (6) and (7), the variables $X_1$, $X_2$, and $X_3$ are included in the regression equation. In Equation (7), variable $X_4$ did not step into the regression analysis. Among these variables, $X_1$ and $X_3$ are positively correlated with $Y$, while, the variables $X_2$ and $X_5$ are negatively correlated with $Y$. The $R^2$ in Equations (6)–(8) are 0.420, 0.544, and 0.675, respectively.

As shown in Table 2, the variables $X_1$, $X_2$ and $X_3$ in the 5-min and 10-min scenarios are significant at the level of $P = 0.05$, while the variables $X_4$ and $X_5$ are not included in the equations. This indicates that the capacity of nursing home, travel costs, and the number of the elderly, are the main factors that affect the $SE_i$ of $i$th subdistrict in both the 5-min and 10-min scenarios. Increasing the number of beds and the search range can significantly improve the spatial equity of a facility, while increasing the number of elderly people will reduce the spatial equity in subdistrict $i$. However, the number of nursing homes and the competition factor $V_j$ have no significant effect on the spatial equity in both 5-min and 10-min scenarios. In the 15-min scenario, variable $X_4$ still is not included in the equation, while the competition factor $V_j$ begins to significantly affect spatial equity. This indicates that, with the increase of spatial scale, the competition factor gradually starts to work. On a micro-spatial scale (such as the 5-min scenario), the distribution of elderly population in neighboring regions has no significant influence on the spatial equity, while, on a macro-spatial scale (such as the 15-min scenario), the competition among people in $i$th subdistrict and its neighboring subdistricts for the limited services must be taken into account.

**Table 2.** Summary of regression model parameters in different scenarios.

| Scenarios | Variables | Coefficients | Std. Error | t | Sig. |
|---|---|---|---|---|---|
| | Constant | 60.109 | 15.249 | 3.942 | 0.000 |
| | $X_1$ | 0.045 | 0.009 | 4.959 | 0.000 |
| 5-min | $X_2$ | −0.010 | 0.003 | −3.270 | 0.002 |
| ($R^2 = 0.420$) | $X_3$ | 2.735 | 0.855 | 3.198 | 0.002 |
| | Constant | 124.053 | 20.104 | 6.171 | 0.000 |
| | $X_1$ | 0.054 | 0.009 | 5.665 | 0.000 |
| 10-min | $X_2$ | −0.018 | 0.004 | −4.907 | 0.000 |
| ($R^2 = 0.544$) | $X_3$ | 2.743 | 0.541 | 5.074 | 0.000 |
| | Constant | 208.239 | 28.630 | 7.279 | 0.000 |
| | $X_1$ | 0.073 | 0.010 | 7.092 | 0.000 |
| 15-min | $X_2$ | −0.034 | 0.005 | −7.057 | 0.000 |
| ($R^2 = 0.675$) | $X_3$ | 2.924 | 0.634 | 4.613 | 0.000 |
| | $X_5$ | −15036.260 | 5008.272 | −3.002 | 0.004 |

### 3.3. The Spatial Patterns of $SE_i$ in Different Scenarios

As shown in Figure 5, in the walking mode, the $SE_i$ in each subdistrict does not increase significantly with time, there are three high spatial equity value clusters around the urban core. In contrast, the spatial equity value in almost all of the urban fringes and urban core areas are low, which indicates that urban fringes and urban core areas are at a disadvantage to nursing homes under walking mode. Residents within these regions cannot access nursing homes easily within walking distance. In the public transit and car modes, the $SE_i$ in each unit increases significantly with time, especially for the subdistricts in the southwest and southeast urban fringes. Southwest urban fringes become the areas with the highest $SE_i$ under public transit and car modes, which indicates that, although these areas have advantages over other regions in terms of accessing nursing home services, this superiority can only be shown under public transit or car mode scenarios. Except for the southwest and southeast urban fringes, the $SE_i$ in other urban fringes and urban core areas are still low, and residents within these areas cannot easily access nursing home services.

Although the maps of spatial equity in different scenarios are varied, the spatial pattern of $SE_i$ in 5-min, 10-min and 15-min integrated scenarios are very similar with one another. As shown in Figure 5, if one does not take southwest and southeast urban fringes into account, the geography of spatial equity of nursing homes in Changchun shows a noticeable ring structure. Specifically, from the urban core to the periphery, the spatial equity presents a "low-high-low" ring structure. The spatial equity value within the urban core and at the periphery is lower than that in the middle area. From the perspective of spatial statistics, we conduct hot spot analysis (*Getis-Ord Gi\**) in the 5-min, 10-min, and 15-min scenarios, respectively, as shown in Figure 6a. Two cold spots and one hot spot are displayed in the 5-min scenario. The two cold spots are located in the urban core and south urban fringe, which indicates that these regions are areas with low $SE_i$. The hot spots are mainly located in the southwest urban fringes, which shows that these areas have a relatively high $SE_i$. In Figure 6b, in the 10-min scenarios, the cold spots are more concentrated in the urban core, and a new hot spot emerges to the east of the urban core, while the southwest urban fringes are still hot spots. The spatial pattern in the 15 min scenario is very similar to that in the 10-min scenario, and the hot and cold spots become more concentrated.

From the above analysis, we can conclude that the geography of spatial equity in different scenarios presents similar spatial patterns, namely, from the urban core to the periphery, the spatial equity presents a "low-high-low" ring structure, and the spatial equity value within the urban core and at the periphery is lower than that in the middle area. From the perspective of spatial statistics, the southwest urban fringes and east urban core are hot spots with higher $SE_i$, while the urban core areas are cold spots with lower $SE_i$, and the other regions belong to non-significant areas.

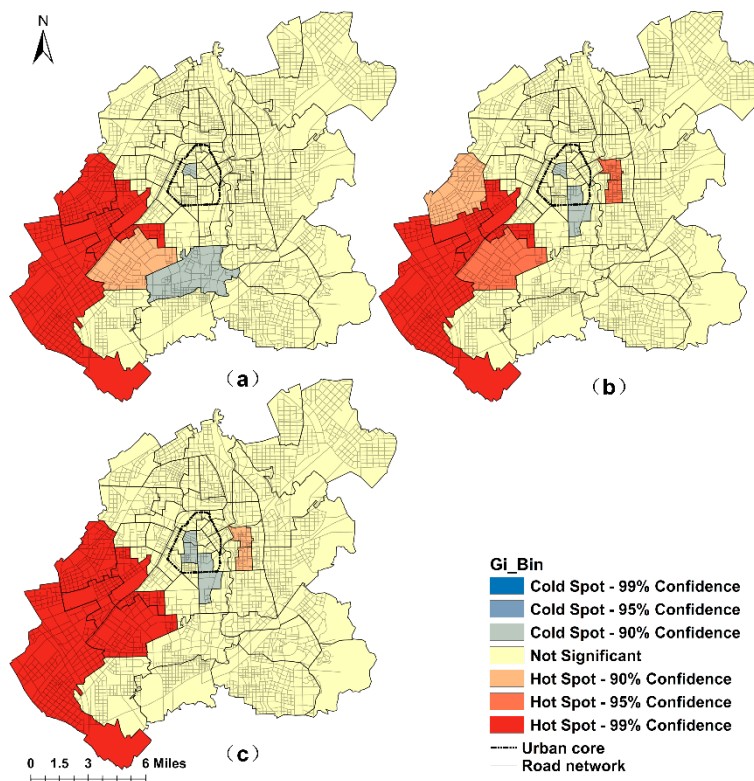

**Figure 6.** Spatial patterns of $SE_i$ in different scenarios. (**a**) 5-min scenario; (**b**) 10-min scenario; (**c**) 15-min scenario.

## 4. Conclusions and Discussions

### 4.1. Conclusions

Trip modes have significant influence on spatial equity of nursing homes, and the geography of spatial equity varied with trip modes. The $SE_i$ under car mode is significantly higher than that in bus and walking scenarios, whereas the $SE_i$ under walking mode is extremely low. In walking mode, areas surrounding the urban core have higher $SE_i$ than other regions, while in public transit and car modes, the southwestern areas have superiority over other regions.

Overall, the $SE_i$ in Changchun is low, and most areas have very low access to nursing homes. In the 5-min integrated scenarios, 40 subdistricts (69%) belong to underserved areas. In the 10-min scenarios, almost 60% of the subdistricts of Changchun have low access to nursing homes. Even in the 15-min integrated situation, there are still more than half of the subdistricts (52%) belonging to the underserved areas. These underserved subdistricts are mainly located in the urban core areas and most of the urban fringes.

Service capacity of nursing home, travel costs, and the number of seniors, are the common factors that significantly affect spatial equity in all three scenarios. The competition factor, namely, the number of seniors in surrounding subdistricts, only affects the 15-min scenario. However, the number of nursing home has no significant effect on the spatial equity of nursing homes.

Although the geography of spatial equity varies with trip modes and scenarios, the geography of spatial equity in different scenarios presents a similar "low-high-low" ring structure; namely, the $SE_i$ within the urban core and at the periphery (except for southwest and southeast urban fringes) is lower than that in the intermediate areas. From the perspective of spatial statistics, southwest urban fringes are hot spot areas with high spatial equity value, whereas the urban core areas are cold spots with low spatial equity value.

*4.2. Discussion*

From a theoretical perspective, the early studies on accessibility were carried out in a single catchment environment [13,44] or in the absence of trip modes and assumed that all residents used the same way to access services [45,46]. But, in reality, not all people adopt the same trip mode when they access service facilities. People of different social classes rely on different trip modes in their daily life, so a multi-trip modes accessibility model can also reflect social equity [15,17]. This paper found that the landscapes of spatial equity vary with trip modes and scenarios. Thus, it is necessary to take trip modes and spatial-temporal scales into consideration when measuring spatial equity. The original spatial equity mode developed by Chang (2011) neglected the spatial competition of people among neighboring spatial units for the limited resources [6]. This paper found that the spatial competition coming from the neighboring regions also affect spatial equity, so the competition factor, especially in a macro-spatial scale, matters. In this paper, based on the multi-mode spatial equity model developed by Chang (2011), we further modified this spatial equity model by integrating spatial competition among consumers in neighboring subdistricts for the limited resources and the attraction factor which was represented by the amounts of amenities (physical therapy equipment, fitness and recreational facilities) equipped in each nursing home into the spatial equity model. We then explored the spatial equity of nursing homes on different spatial scales. This research can provide some improvement to current research that does not consider the impact of trip modes and spatial scales on accessibility.

From a policy perspective, this research found that service capacity, travel costs, and the number of seniors, are the main factors influencing spatial equity. However, the number of nursing homes has no significant effect on the area's spatial equity, which indicates that it is service capacity—not the number of nursing homes—that is the key to the spatial equity of nursing home resources. In addition, the "ring structure" and big spatial variance of spatial equity are highly related to the varied capacities and location preferences of different nursing homes. Because larger nursing homes always need more space to provide more services, they tend to be located in the intermediate zones between the urban core and urban fringe, while small nursing homes tend to be located in urban core areas. These findings have important policy implications to the spatial planning of nursing homes. In the spatial planning of nursing homes, it is necessary to focus on the improvement of service capacity rather than the blind increase of the number of nursing homes. What is more, it is also necessary to consider the balanced distribution of different types of nursing homes; the urban core areas are densely populated areas, but due to the scarce land provision and expensive rent, the capacities of nursing homes are limited. On the contrary, because the urban peripheries are sparsely populated areas with enough space and low land rent, they are the ideal locations for larger nursing homes. To avoid the spatial mismatch between the elderly and the nursing homes, it is necessary to consider the spatial equilibrium of nursing homes with different capacities.

This paper does have some limitations. First, factors seniors taking into account when choosing pension services are quite complex; for example, the socio-economic status of the elderly, the quality and costs of the nursing homes, etc. Due to privacy issues, some data such as the service fee and the service quality of each nursing homes cannot be collected currently, so we cannot include those factors into our analysis, we can only use the amount of the physical therapy equipment, fitness and recreational facilities to represent the attraction of each nursing home. Secondly, spatial equity was determined by both spatial factors and non-spatial factors. The spatial equity model used in this paper is mainly to address the spatial factors. How to integrate more non-spatial factors, such as individual preference, income, age, health status, into traditional spatial equity modes can be the focus of our future study.

**Author Contributions:** Shuju Hu, Wei Song and Chenggu Li conceived and designed the study; Shuju Hu performed data analysis; Shuju Hu, Wei Song and Jia Lu analyzed the results, wrote and revised the manuscript. All authors have read and approved the final manuscript.

**Funding:** This study was supported by the National Natural Science Foundation of China (Grant No.41871158).

**Acknowledgments:** The authors would like to thank the four anonymous reviewers' comments and managing editor's highly efficient work in processing our manuscript. We also thank Catherine Lange from the Writing Center of University of Louisville for her suggestions to improve the language of the manuscript. Finally, we are grateful for the financial support from China Scholarship Council.

**Conflicts of Interest:** The authors declare no conflict of interest.

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
