# Peer review of "The Spatial Equity of Nursing Homes in Changchun: A Multi-Trip Modes Analysis"

_ijgi, doi:10.3390/ijgi8050223_

Round 1
Reviewer 1 Report
1.In this paper, the concept "spatial equity" was changed to "accessibility"?what is the difference between the tow concepts and if they are the same concept or issue?
2.How can the author draw the conclusion:“However, few scholars value the differences of various distance measurement methods on the evaluation of accessibility.”(Line 71-73)。In my opinion ,many scholars have do many research jobs in the field of accessibility assessment about urban green space, disaster shelter, sports facilities, hospital, bank, and so on.
3.Specifically, what is the " improved potential model", the difference to the "traditional model" should be explained in detail.
4.How the data in the network analysis were organized? which parameters have been taken into count to the network?
5.As to Figure 2, the legend should be created based on the symbol system of the features in the map.
6.English language needs further improvement
Author Response
Dear, reviewer:
Thank you for your comments concerning our manuscript. These comments are all valuable and very helpful for revising and improving our paper. We have studied comments carefully and have made the corrections. We hope it meets with approval.

Reviewer 2 Report
@page { margin: 2cm } p { margin-bottom: 0.25cm; line-height: 120% }The article presents an analysis of the spatial equity of nursing home in Changchun, China. The spatial equity in this article is analyzed according to the location of the nursing homes, the characteristics of each of them (number of beds and amenities), elderly population, and the trip modes (walking, car, and bus).
The article is well-written and the analysis is interesting. However, I have some concerns about the technical part of the article. I consider that the methodology applied for analyzing the spatial equity should be fully described together with the real contributions of the article. It is not clear if the contribution is the study itself (with the results), the improvements of the spatial equity analysis, or both. For example, if the contribution is the way the spatial equity is analyzed, I think that authors must include a methodology considering reproducibility of the steps and activities applied. In Section 2 authors describe the methodology without denoting aspects such as: the way all data (amenities, elderly population, travel costs, etc.) have been obtained or extracted, related methodologies of spatial equity analysis, improvements about the methodologies based on spatial equity, etc. Section 2.1 only shows the location of nursing homes and the number of beds, while amenities, populations, trip modes, and costs are absent. I consider that authors must include a methodology about the way the study was performed, denoting hypotheses, analysis setup, sources, and finally results. A more complete related works is also required about other studies analyzing spatial equity (also in other contexts) in the literature.
Finally, with respect to the results, I think that some conclusions must be more explained in order to understand how they have been obtained from the analysis. For instance, the sentence “A higher SE implies a richer pension and a higher accessibility”, why a richer pension? Was the pension considered in the formula?. Another one is: “However, the number of facilities has no significant effect on the spatial equity of nursing homes”, why? Where we can see this conclusion?
Author Response

(The authors gave the same response as above.)

Reviewer 3 Report
The authors have done good work, and the paper is of excellent scientific quality. Please get this paper reviewed either by a native English speaker or a professional editing company for grammar, and it should be publishable in a good journal.
Author Response

(The authors gave the same response as above.)

Reviewer 4 Report
Develop a solid argument for the methods. While the authors explained various methods, they failed to address (1) which methods were selected for this study, and (2) why those were better/ more effective than others.
Is it assumed that across all spatial units, the modal share between walking, public transit, and cars is constant (i.e. 0.302 vs. 0.397 vs. 0.19)? Otherwise, a brief descriptive stat for variables, including min, max, mean and median for each parameter should help. What are the variations across these 58 spatial units?
Also, it is unclear what is the origin – destination in this accessibility model? Did the authors use the centroid of each spatial unit as the origin to nursing homes as destinations? S
Where is the explanations on how these equations were processed? In other words, which software was used for statistical treatments, mapping, etc?
Lastly, the authors argued for the need of incorporating social contexts – i.e. different populations or sub-populations, it doesn’t appear to me that the authors addressed any of the social factors (income, access to different travel modes, etc.) and the effects of the target populations in this study. For that matter, the authors should not argue this study, or approach, improved the traditional accessibility model. In the discussion, the authors pointed out the privacy as the reason for the lack of socio-economic factors; however, since it is aggregated to 58 spatial units, if available, the authors should certainly be able to incorporate such information for actual “improvement” of accessibility mode for spatial equity.
Specifically,
Lines 48-70: Need to validate the models: the authors kept arguing the methods were “most” used or “most” valid, and yet there was no valid comparison of these methods to others for their superiorities.
Lines 85-99: Address the findings from previous studies on multimodal effects on spatial equity. For example, since the study explores walking vs. bus vs. car, at least the authors should address the observed differences between non-motorized (walking and/or biking) and motorized (public transit, automobile, etc.) on accessibility.
Lines 100-109: I suggest that the authors need to develop the research questions with more specifics, particularly on different modes and different scenarios. What are these scenarios? Bring the brief explanation upfront to help the readers as well as the previous works on those scenarios – to validate the scenario selection.
Line 113: what are these sub-spatial units in the city? Are they equivalent to any US spatial units, for example census tracts? block groups? Or blocks?
Lines 114-118: is there any other information – besides the number of beds – important to address/ examine to understand the spatial equity? Or explain why three categories – by the # of beds – are important to implement. Specifically, for lines 149-153, the authors mentioned this element as Sj. Thus, explain what ‘amenities’ were taken into account to make up S.
Lines 139-165: explain equation (3) first and better before adding equation (4). Since travel cost is the indispensable element in the model, explain the relationship between population (p), portion of the population (a), and the travel times (T).
Lines 189-200: explain better. What did the authors mean by “5-minthe (minutes I assume from line 190)average value of spatial equity of 58 spatial units in 5-min is 40.2”?
Lines 204-226: the authors need to explain the general equation for regression equations 5-7 beforehand. I would suggested to check the equations and the explanations thoroughly. For example, in the line 221, X4 is not selected in the model, but appears in the equation 7.
Lines 253-262. Please explain Getis-Ord Gi* with more details – since the authors didn’t mention the method anywhere in method. Also, explain why and how this method helped the authors illustrate the geography of spatial equity. What did the authors mean by hot spots and cold spots? This needs a further explanation for spatial clustering.
Author Response

(The authors gave the same response as above.)

Round 2
Reviewer 2 Report
I think that authors have considered all my revisions/suggestions, so the the article is accepted.
Reviewer 4 Report
Thanks for reviewing my feedbacks thoroughly, and having them reflected in the paper.